# Multiview Human Body Reconstruction from Uncalibrated Cameras

**Zhixuan Yu**
University of Minnesota
yu000064@umn.edu

**Linguang Zhang**
Meta Reality Labs
linguang@meta.com

**Yuanlu Xu**
Meta Reality Labs Research
yuanluxu@meta.com

**Chengcheng Tang**
Meta Reality Labs
chengcheng.tang@meta.com

**Luan Tran**
Meta Reality Labs
tranluan07@meta.com

**Cem Keskin**
Meta Reality Labs
cemkeskin@meta.com

**Hyun Soo Park**
University of Minnesota
hspark@umn.edu

## Abstract

We present a new method to reconstruct 3D human body pose and shape by fusing visual features from multiview images captured by uncalibrated cameras. Existing multiview approaches often use spatial camera calibration (intrinsic and extrinsic parameters) to geometrically align and fuse visual features. Despite remarkable performances, the requirement of camera calibration restricted their applicability to real-world scenarios, *e.g.*, reconstruction from social videos with wide-baseline cameras. We address this challenge by leveraging the commonly observed human body as a semantic calibration target, which eliminates the requirement of camera calibration. Specifically, we map per-pixel image features to a canonical body surface coordinate system agnostic to views and poses using dense keypoints (correspondences). This feature mapping allows us to semantically, instead of geometrically, align and fuse visual features from multiview images. We learn a self-attention mechanism to reason about the confidence of visual features across and within views. With fused visual features, a regressor is learned to predict the parameters of a body model. We demonstrate that our calibration-free multiview fusion method reliably reconstructs 3D body pose and shape, outperforming state-of-the-art single view methods with post-hoc multiview fusion, particularly in the presence of non-trivial occlusion, and showing comparable accuracy to multiview methods that require calibration.

## 1   Introduction

Cameras are an integral part of our lives for us to capture and share priceless moments. In particular, social videos voluntarily captured by multiple viewers watching the same scene, *e.g.*, friends recording a street busker simultaneously, provide a new form of popular content for visual communication in social media.

These videos possess two properties. (1) The videos are, by nature, multiview from social members, which provide redundant yet distinctive visual information to model the 3D body geometry of a target subject. For instance, the busker's right elbow is occluded by his torso in one view but visible from another viewpoint as shown in Figure 1(a) [2]. This self-occlusion can be reasoned by consolidating multiview information. (2) These videos are, in general, not spatially calibrated (intrinsic and extrinsic parameters). There exists neither a calibration pattern nor common visual features to register 3D

36th Conference on Neural Information Processing Systems (NeurIPS 2022).

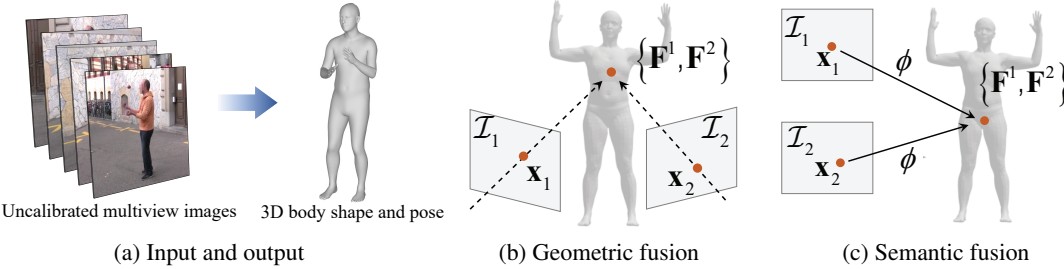

| (a) Input and output | (b) Geometric fusion | (c) Semantic fusion |

Figure 1: (a) We present a new method to reconstruct the 3D human body shape and pose by fusing visual features from uncalibrated multiview images. All results in this manuscript are generated before blurring the face of the subject. (b) Fusing visual features $\{\mathbf{F}^1, \mathbf{F}^2\}$ from multiview images require camera calibration to align features. (c) Instead, we use body semantics, e.g., dense keypoints, to align features, eliminating the requirement of camera calibration.

camera poses (*e.g.*, structure from motion) because of wide-baseline cameras. Existing computer vision solutions leveraging multiview calibration [14, 16, 13, 39, 49] are therefore not applicable. On the other hand, single view approaches [21, 23, 47, 18] for 3D body reconstruction can be combined in a post-hoc way, *e.g.*, taking the average of global visual features, which shows limited performance.

In this paper, we propose a novel method to reconstruct 3D human pose and shape from uncalibrated multiview images. We use the commonly observed human body as a *semantic calibration target* to align visual features across views: the dense keypoints (correspondences) [11] detected in an image provide an injective mapping from image coordinates to the canonical body surface coordinates regardless of views and poses, which allows semantically aligning and then, fusing visual features from multiview images.

We integrate the following desired properties to learn visual features for reconstructing 3D body pose and shape. (1) View independence: Without camera calibration, in order to learn and fuse visual features independent of views, we need to transport them to a shared space. To achieve this, we semantically, instead of geometrically, transfer and align visual features based on dense keypoint mapping to the body surface. Such a unified coordinate system makes visual feature extraction agnostic to views and poses. (2) Locality: We hypothesize that a local visual feature for each pixel can describe the fine-grained relationship between human body parts around the pixel. In conjunction with global features that describe a holistic pose, we learn local features from multiview images, jointly. (3) Confidence: Views are not equally informative and can complement each other in inferring different body parts. We design an attention mechanism to reason about the confidence of visual features, both globally across multiple views and locally within each view. Across multiple input images, a higher global confidence score is assigned to a view generating more accurate reconstruction, *e.g.*, a less occluded one. Within each view, higher local confidence scores are assigned for body parts that are easier to infer, *e.g.*, a clearly visible limb.

We design an end-to-end network that learns to reconstruct 3D body geometry from multiview images. It takes as input, a set of multiview images and extracts per-pixel features and their confidences. We use dense keypoint mapping to semantically align the features in the canonical coordinate system defined by the human body surface. These aligned features from multiview images are fused by using the predicted confidences, which form unified per-vertex features. We concatenate those fused per-vertex features together with image feature for learning body model parameters.

We apply our method to reconstruct the 3D human body from uncalibrated multiview images. Our method can effectively fuse visual features from multiview images, outperforming single view approaches with post-hoc fusions, in particular, for self-occlusion, and showing comparable performance compared to multiview approaches that use calibrated images. We demonstrate our work in realworld scenarios such as social videos where camera calibration is challenging.

**Contributions** This paper makes three major technical contributions. (1) We propose a novel multiview method for human pose and shape reconstruction that scales up to an arbitrary number of uncalibrated camera views (including single view), guided by dense keypoints. (2) We introduce a novel self-attention based multiview feature fusion method that takes uncertainties across and within different views into consideration. (3) We achieve state-of-the-art results among calibration-free 3D human body pose and shape reconstruction approaches.

## 2   Related Work

To handle the challenge due to occlusion, single-view-based methods infer the occluded parts in a data-driven manner, while multiview approaches can fuse visual features from different views.

**Single View 3D Human Reconstruction** Reconstructing occluded 3D content from a single view image is a geometrically ill-posed problem. With some assumptions about scenes, it is possible to reason about the occluded geometry. For instance, for humans, a low dimensional parametric model such as SMPL [32] can be used to reconstruct shape and pose by matching 2D sparse landmarks [9, 3] and semantic segmentation masks [26]. On the other hand, learning-based approaches [30, 22] directly regress the model parameters from an image [21], or semantic representations such as 2D keypoint heatmaps and silhouettes [37], part segmentation [35], and dense correspondences [47]. Such learning-based approaches rely on datasets with 3D ground truth [15, 20, 33, 50] to learn the relationship between an image and the shape and pose, which is prone to overfitting when applying to a novel scene image. To address the limited 3D ground truth data, self-supervised learning can be used by generating photorealistic data [23] and by enforcing consistency for predictions from multiple views [36, 48]. To enhance the expressibility of the model, the parametric representation is relaxed by incorporating a volumetric representation [45, 17], a point cloud [8], or vertex locations of a pre-defined mesh [24, 52]. Compared with single view reconstruction, our approach is tailored for multiview feature fusion that significantly improves the performance.

**Multiview Fusion for 3D Human Reconstruction** When camera calibration (intrinsics and extrinsics) is available, single view reconstruction methods can be extend to multiview settings. For instance, SMPLify [3] is extended to reconstruct a 3D body geometry in an unified coordinate system where its validity can be measured by projecting onto multiview images, *i.e.*, the reprojection error of the 2D keypoints and silhouette is minimized to learn a geometrically coherent model [28]. Multiview geometry offers a geometric way to fuse multiview features for self-occlusion reasoning [55, 57]. For instance, visual features [13] or predictions [56] can be transferred to other views through epipolar lines, inverse projection [16, 44], or a latent view-invariant space [39].

When the camera calibration is not available, the learning-based approaches can be used to fuse multiview features. For example, a fusion module can be learned to warp a feature map from one view to align with that of another view [38, 46], which does not rely on calibration but requires learning a large number of parameters to model the cross view geometry. Besides, uncalibrated multiview images can be used in a sequential manner. For example, a recurrent framework can be used to progressively refine an initial single view prediction of camera and human model parameters, view by view and stage by stage [29]. The requirement of synchronization between cameras can be relaxed by learning a probabilistic distribution of the model parameters [42]. Unlike existing work, we fuse multiview images without calibration using the observed human body as a semantic calibration target. The features are transported to the canonical body surface where a self-attention module is used to combine multiview features effectively.

**Dense Keypoints for 3D Human Reconstruction** Dense keypoint estimation, *e.g.* DensePose [11], establishes correspondences between image pixels and the canonical body surface coordinate system. This provides a strong cue to infer 3D body pose and shape from a single image that alleviates the self-occlusion problem [40]. Due to such a desirable property, single view approaches employ dense correspondence for supervision [10, 52] or as an additional cue [53, 47]. Further, it can be used to reconstruct 3D body pose and shape from multiview images [27, 51]. Inspired by DecoMR [52], we leverage dense correspondences to re-arrange visual features from different images into a canonical body surface, which allows us to effectively fuse multiview features. Note that DecoMR [52] is a single view method that transfers features to a 2D canonical body surface coordinate and regress a discrete view-specific vertex location map; while ours transfer features from an arbitrary number of views to a continuous 3D canonical body mesh for fusion and regressing a view-indepdent body model.

## 3   Method

We present a novel method that reconstructs 3D human body pose and shape from an arbitrary number of uncalibrated multiview images. We leverage dense keypoints to map visual features from multiview images to a common canonical coordinate system, which allows effectively fusing them for 3D reconstruction. Figure 2 shows the overview of our pipeline.

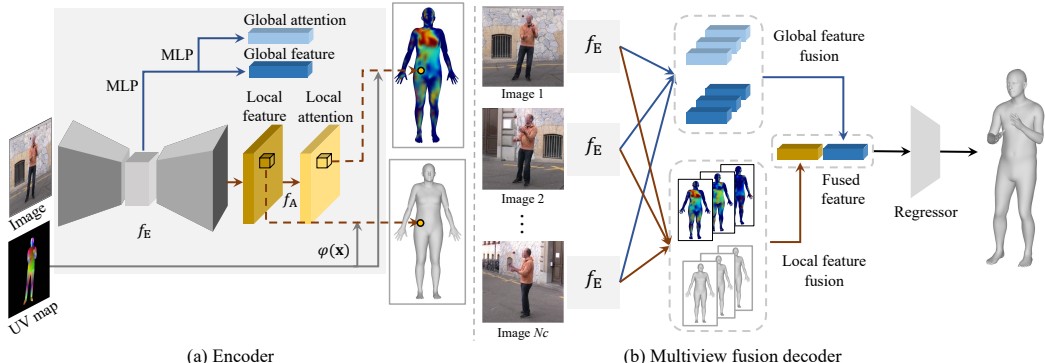

(a) Encoder     (b) Multiview fusion decoder

Figure 2: (a) We design a new neural architecture for the feature encoder $f_{\mathrm{E}}$ that outputs visual features for 3D reconstruction. The global feature is computed from the bottleneck feature, and its global attention is obtained by transforming the global feature through fully connected layers. The encoder also output per-pixel (local) features and their attention that are mapped to body vertex coordinates via dense keypoints (UV map). (b) We combine the global and local features from the multiview images using their attention. The fused local features and global features are concatenated and feed into a MLP regressor for predicting 3D shape and pose.

## 3.1 Human Body as Semantic Calibration Target

Given an image $\mathcal{I} \in [0,1]^{W \times H \times 3}$ containing a person with width $W$ and height $H$, we aim to recover the person's 3D body pose and shape represented as mesh vertices $\mathbf{Z} = \{\mathbf{z}_i \in \mathbb{R}^3\}_{i=1}^{N_v}$ by estimating SMPL model [32] parameters $\Theta$.

This problem in general can be formulated as a composition of an image encoder and a mesh decoder:

$$\mathbf{F} = f_{\mathrm{E}}(\mathcal{I}; \theta_E), \quad \mathbf{Z} = f_{\mathrm{D}}(\mathbf{F}; \theta_D), \tag{1}$$

where $f_{\mathrm{E}}(\mathcal{I}; \theta_E)$ denotes an image feature encoder that extracts feature maps $\mathbf{F}$ from the input image $\mathcal{I}$, and $f_{\mathrm{D}}(\cdot; \theta_D)$ denotes the mesh decoder that takes a set of the image features to reconstruct the 3D human pose and shape.

Existing multiview body pose and shape reconstruction methods extend single view methods by leveraging camera calibration to fuse multiview image information [13, 16]. In contrast, we leverage the observed human pose as a semantic cue to fuse visual features from multiview images. Consider a discrete mapping $\phi(\mathbf{x}, \mathcal{I})$ that maps a pixel in an image $\mathcal{I}(\mathbf{x})$ to a vertex on the 3D body surface, $\mathbf{z}_i$. This offers a new way to align visual features in a common vertex coordinate system across views:

$$\mathbf{F}_i = f_{\mathrm{E}}(\mathbf{x}, \mathcal{I}; \theta_E), \quad \text{where} \quad i = \phi(\mathbf{x}, \mathcal{I}), \tag{2}$$

and the encoder $f_{\mathrm{E}}(\cdot)$ learns the corresponding image features $\mathbf{F}_i$ for a certain body vertex $\mathbf{z}_i$. We assume the 3D body vertex index $i$ are consistent across people, views, and poses.

Figure 1(b,c) illustrates the comparison between geometric fusion that relies on camera calibration to align visual features from multiview images via epipolar line [13] or triangulation [16]. In contrast, our approach uses mappings from image points to 3D body vertices, $\phi(\mathbf{x}, \mathcal{I})$, to semantically align features without camera calibration.

## 3.2 Local Feature Registration

A key challenge of learning visual features from Equation (2) lies in the non-differentiability of the index map $\phi$. The domain (pixel coordinates $\mathbf{x}$) is not continuous, and any $\mathbf{x}$ that does not map to an integer index is undefined. We address this challenge by relaxing the discrete domain, *i.e.*, representing the feature $\mathbf{F}_i$ using a set of continuous vicinity features from a continuous mapping.

Using a dense (continuous) keypoint map $\psi(\mathbf{x}, \mathcal{I})$ that maps pixel coordinates $\mathbf{x} \in \mathbb{R}^2$ to continuous UV coordinates $\mathbf{u} \in \mathbb{R}^2$, we define the feature of the $i$-th body vertex $\bar{\mathbf{F}}_i$ as a weighted sum of all features nearby in the UV space (Figure 3):

$$\bar{\mathbf{F}}_i = \frac{\sum_{\mathbf{u}_j \in \mathcal{N}_i} w_j f_{\mathrm{E}}(\psi^{-1}(\mathbf{u}_j), \mathcal{I}; \theta_E)}{\sum_{\mathbf{u}_j \in \mathcal{N}_i} w_j}. \tag{3}$$

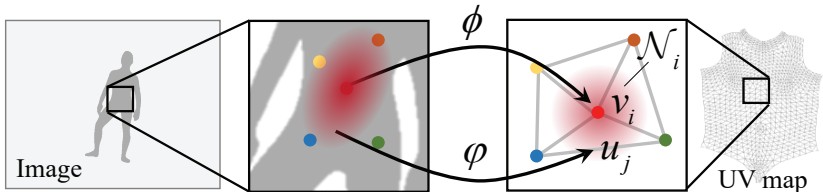

Figure 3: We approximate the discrete feature mapping $\phi$ using a continuous mapping $\psi$, i.e., the feature in the $i$-th vertex is approximated by the weighted average of the nearby features, which enables differentiable feature learning.

Here, $\mathcal{N}_i$ is a set of neighboring locations of the $i$-th vertex for which the inverse are well-defined pixel coordinates; $w_j$ is the weight for the visual feature corresponding to UV coordinates $\mathbf{u}_j$. Given the location of the $i$-th vertex in the UV coordinate system $\mathbf{v}_i$, its neighborhood is defined as:

$$\mathcal{N}_i = \{\mathbf{u}_j : d(\mathbf{u}_j, \mathbf{v}_i) < \tau, \ \psi^{-1}(\mathbf{u}_j) \in \Lambda_{\mathcal{I}}\}, \tag{4}$$

where $d(\cdot, \cdot)$ measures the geodesic distance between two points in the 3D body surface and $\Lambda_{\mathcal{I}}$ is the lattice of the input image. We further define the weight by applying an RBF kernel on the geodesic distance, *i.e.*,

$$w_j = \exp\left(-\frac{d(\mathbf{u}_j, \mathbf{v}_i)^2}{2\sigma^2}\right), \tag{5}$$

where $w_j$ is the weight for the $j$-th neighbor $\mathbf{u}_j$, and $\sigma$ controls the effective range of the neighbors. To avoid numerical instability, in practice, we assign zero for features when $\sum_{\mathbf{u}_j \in \mathcal{N}_i} w_j < 10^{-6}$.

### 3.3 Multiview Feature Fusion via Self-attention

Depending on the camera configuration and the human pose, some views can be more informative for reconstructing a certain body area than the others which suffer from occlusion or depth ambiguity. We address this by a self-attention mechanism that learns to assign a weight for each view at different body areas.

With Equation (2), visual features from multiview images can be aligned with respect to the 3D body surface indices, which allows us to fuse the image features from multiview images:

$$\bar{\mathbf{F}}_i^* = \sum_{c=1}^{N_c} \alpha_i^c \bar{\mathbf{F}}_i^c, \tag{6}$$

where $\bar{\mathbf{F}}_i^*$ is the $i$-th fused image feature and $\alpha_i^c$ is the weight indicating the feature confidence for the $c$-th camera view, respectively.

We represent the feature confidence $\alpha_i^c$ as a self attention, and it can be learned in an unsupervised way:

$$a_c(\mathbf{u}) = f_{\mathrm{A}}(f_{\mathrm{E}}(\psi^{-1}(\mathbf{u}), \mathcal{I}_c; \theta_E); \theta_A), \tag{7}$$

where $a_c(\mathbf{u})$ is the attention score at the UV coordinates $\mathbf{u}$ from the $c$-th view, and $f_{\mathrm{A}}(\cdot; \theta_A)$ is the learned function that predicts the attention given the set of features from the $c$-th view, respectively.

Similar to Equation (3), we represent the feature confidence as a weighted sum of attention scores from neighboring locations:

$$\alpha_i^c = \frac{\exp(\beta_i^c)}{\sum_{k=1}^{N_c} \exp(\beta_i^k)}, \quad \text{where} \quad \beta_i^c = \frac{\sum_{\mathbf{u}_j \in \mathcal{N}_i} w_j^c a_c(\mathbf{u}_j)}{\sum_{\mathbf{u}_j \in \mathcal{N}_i} w_j^c}, \tag{8}$$

and $w_j^c$ is the weight for the $j$-th neighbor, $\mathbf{u}_j$, in the UV-coordinate system, as defined in Equation (4-5). We apply the softmax operation to normalize feature confidence across views. Figure 4 illustrates the transferred learned attention values on the body mesh. Overall, the learned attention scores are consistent with our intuition about contributions to body regions from different camera views.

While per-pixel features are useful for capturing fine-grained geometry, the holistic shape such as pose can be better represented by a global feature. In conjunction with per-pixel features, we incorporate

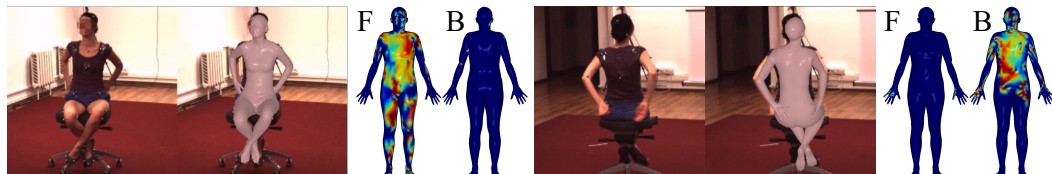

Figure 4: We use a self-attention mechanism to fuse the local and global features across views. The intensity in the heatmaps (right) indicates the attention predicted by the images (left). "F" denotes the front view and "B" denotes the back view. The attention is expected to highly correlated with the visible regions of the body because the features can be confidently predicted from the encoder. The reconstructed 3D mesh overlays the input image.

global features. To fuse global features across views, we utilize a self-attention mechanism similar to local feature fusion with minor modifications. Instead of learning a single score for each view, we learn an attention score for each channel of the global feature from each view and then normalize it to be used for combining global features across views.

### 3.4 Loss and Network Design

Given the neural network output $\boldsymbol{\Theta}$ and subsequent $\mathbf{Z}$, we learn our model in an end-to-end manner by minimizing the following loss:

$$\mathcal{L} = \sum_{\widehat{\boldsymbol{\Theta}} \in D_\mathrm{V}} \|\boldsymbol{\Theta} - \widehat{\boldsymbol{\Theta}}\|_2 + \lambda_\mathrm{P} \sum_{\widehat{\mathbf{P}}_{3\mathrm{D}} \in D_P} \|J\mathbf{Z} - \widehat{\mathbf{P}}_{3\mathrm{D}}\|_2^2 + \lambda_\mathrm{K} \sum_{\widehat{\mathbf{P}}_{2\mathrm{D}} \in D_\mathrm{K}} \|\Pi(J\mathbf{Z}) - \widehat{\mathbf{P}}_{2\mathrm{D}}\|_2^2,, \tag{9}$$

where three terms are the losses for model parameters, 3D joint error, and 2D joint error, respectively, and $\lambda_\mathrm{P}$ and $\lambda_\mathrm{K}$ balance the loss magnitudes. $D_V$, $D_P$, and $D_K$ are their datasets, *i.e.*, we use multiple datasets to improve generalizability. $\widehat{\boldsymbol{\Theta}} \in \mathbb{R}^{85}$, $\widehat{\mathbf{P}}_{3\mathrm{D}} \in \mathbb{R}^{3M}$, and $\widehat{\mathbf{P}}_{2\mathrm{D}} \in \mathbb{R}^{2M}$ are the ground truth model parameters, 3D vertices, 3D joint positions, and 2D joint positions, with $N$ and $M$ denoting the number of vertices and joints, respectively. $J \in \mathbb{R}^{3M \times 3N}$ is the pre-defined 3D joint regression matrix that linearly maps the 3D vertices to the 3D joints [32], and $\Pi(\cdot)$ is the operation of camera projection that projects the 3D joints to 2D joints in the image plane. $\mathcal{L}_\Theta$ measures the Euclidean error between the ground truth model parameters and predicted ones, $\mathcal{L}_\mathrm{P}$ measures the 3D error of the joints, i.e., $J\mathbf{Z}$ is the $M$ joint locations, and $\mathcal{L}_\mathrm{K}$ measures the 2D projection of the joint locations.

Note that neither the local feature registration nor the multiview fusion assumes a fixed camera configuration or a fixed number of views. We fully leverage this advantage in our training pipeline: at each step, we randomly choose a subset of views from each data sample to train the model. As a result, we are able to mix multiview datasets and single view datasets to train a model that can handle the variable number of views.

### 3.5 Implementation Details

We design the encoder $f_\mathrm{E}(\cdot; \theta_E)$ as a ResNet-50 backbone [12], that takes a $224 \times 224 \times 3$ image as an input and outputs a global feature vector with 256 dimensions and a $56 \times 56$ local feature map with 256 dimensions. The local attention function $f_\mathrm{A}(\cdot; \theta_A)$ is modeled by a sequence of convolutional layers, while the per-channel attention for global feature is modeled by a fully connected layer. Each local feature vector associated with a vertex of a down-sampled SMPL mesh [24] goes through a three-layer MLP and reduces its dimension from 256 to 5 [54], and then we concatenate all 431 of them to form a 2155-dimensional feature vector. This aggregated local feature vector is then concatenated with the global feature vector and fed into a MLP regressor similar to that in HMR [21].

We use a pre-trained DensePose model for $\psi$, which partitions the whole body surface to 24 different parts where UV map is defined for each part. In practice, $d(\mathbf{u}, \mathbf{v})$ in Equation (4) and Equation (5) is set to infinity if $\mathbf{u}$ and $\mathbf{v}$ are from different parts. Another important detail is that for a vertex that corresponds to multiple 2D UV coordinates, we take the minimum when computing the geodesic distance, *i.e.*, replacing $d(\mathbf{u}_j, \mathbf{v}_i)$ with $\min(\{d(\mathbf{u}_j, \mathbf{v}_i^k)\})$ in Equation (4) and Equation (5), where $\mathbf{v}_i^k$ is the $k$-th 2D UV coordinates that the same $i$-th vertex corresponds. In fact, this serves as an approximation of $\phi$ using $\psi$. We set $\tau = 0.05$ and $\sigma = 2.33 \times 10^{-2}$.

# 4 Experiments

We validate our calibration-free multiview fusion approach on multiple datasets varying from indoor to outdoor, controlled to in-the-wild environments. We refer the reader to check additional results, experiments and implementation details in the *supplementary material*.

## 4.1 Datasets

**Human3.6M**[1] [15] is a large-scale multiview dataset with ground truth 3D human pose annotation. We follow the standard training/testing split: using subject S1, S5, S6, S7 and S8 for training, and subject S9 and S11 for testing. We reconstruct the ground truth 3D human mesh in the format of the SMPL [32] model by applying MoSh [31] to the sparse 3D MoCap marker following previous works [21, 24, 52, 51].

**UP-3D** [26] is an in-the-wild single view dataset containing 8,515 images annotated with ground truth 2D keypoints and 3D body mesh. The 3D body mesh is obtained by fitting the SMPL [32] model to images from the Leeds Sports Pose [19], MPII Human Pose [1], and FashionPose [5] datasets. We use the standard training split [24, 52].

**MARCOnI** [6] is a multiview dataset including both indoor and outdoor images for evaluating marker-less motion capture methods. This dataset contains 12 real-world sequences captured by 3-16 cameras with varying modalities from 1-2 subjects each in different scenes, *e.g.*, Soccer, Kickbox, Juggling, etc. We use this dataset to qualitatively evaluate our method and test the generalizability of the model.

**VBR** [2] is a multiview dataset that include challenging social videos captured by multiple members in a scene. The baseline between cameras are wide. It contains three synchronized multiview sequences including Juggler, Magician, and Rothman. We use this dataset to qualitatively evaluate our method and test the generalizability of the model.

**Social Videos**[2] are a new multiview dataset we collected for qualitative study. It features multiple persons performing dynamic group activities, such as playing basketball and name tag ripping, which introduces frequent occlusion and disocclusion. Up to 9 subjects are captured from 4 hand-held cameras in 6 sequences. We run an off-the-shelf 2D human pose detection and tracking tool [7] with manual correction and multiview association to obtain ground truth per-person multiview tracking data.

Note that we run an off-the-shelf structure-from-motion system (COLMAP [41]) to solve the camera parameters for multiview datasets above. However, as expected, for almost all cases the bundle adjustment fails to converge and it reports that no good initial image pair was found, i.e., lack of correspondences.

## 4.2 Metrics

Since the proposed method reconstructs the 3D body shape and pose (mesh) and uses the local features for recovering fine details, we evaluate of the method using Mean Per-Vertex Position Error (MPVPE) and Mean Per-Joint Position Error (MPJPE), both after Procrustes Analysis (PA). MPJPE-PA primarily evaluates the pose estimation accuracy of the model, whereas MPVPE-PA evaluates the accuracy of the reconstructed body shape.

## 4.3 Comparison with State-of-the-art Methods

**Baselines** SMPLify [3] optimize SMPL model parameters by fitting to 2D keypoints. HMR [21] reconstructs the 3D body shape and pose from a monocular image by estimating the SMPL model parameters in an iterative manner. GraphCMR [24] leverages a graph convolutional network to first reconstruct the non-parametric mesh, which is then refined by fitting SMPL model parameters. SPIN [23] further improves the accuracy by combining iterative fitting and regression into a self-improving loop. DecoMR [52], Pose2Mesh [4], I2lMeshnet [34], MeshTransformer [30] and PyMAF [54] are several more recent monocular methods. Multiview SPIN and LVS [43] are two methods that leverage multiview calibration for estimating SMPL model parameters as a reference.

---

[1]Only authors from University of Minnesota downloaded and accessed to Human3.6M dataset. Authors from Meta Reality Labs don't have access to it.

[2]Social Videos dataset is collected at University of Minnesota, not Meta Reality Labs.

| Method | Type | Multiview reconstruction | | | Single view reconstruction | |
|---|---|---|---|---|---|---|
| | | Calibration-free | MPJPE-PA | MPVPE-PA | MPJPE-PA | MPVPE-PA |
| SMPLify [3] | Mono | ✓ | N/A | N/A | 82.3 | N/A |
| HMR [21] | Mono | ✓ | $57.8 \pm 10.7$ | $67.7 \pm 15.4$ | 56.8 | 65.5 |
| GraphCMR [24] | Mono | ✓ | $50.9 \pm 9.1$ | $59.1 \pm 13.4$ | 50.1 | 56.9 |
| SPIN [23] | Mono | ✓ | $44.5 \pm 7.9$ | $51.5 \pm 11.8$ | 41.1 | 49.3 |
| DecoMR [52] | Mono | ✓ | $42.0 \pm 8.8$ | $50.5 \pm 14.1$ | 39.3 | 47.6 |
| Pose2Mesh [4] | Mono | ✓ | N/A | N/A | 47.0 | N/A |
| I2lMeshnet [34] | Mono | ✓ | N/A | N/A | 41.1 | N/A |
| MeshTransformer [30] | Mono | ✓ | N/A | N/A | **36.7** | N/A |
| PyMAF [54] | Mono | ✓ | N/A | N/A | 40.5 | N/A |
| MV-SPIN [43] | Multi | ✗ | 35.4 | N/A | N/A | N/A |
| LVS [43] | Multi | ✗ | **32.5** | N/A | N/A | N/A |
| Liang [29] | Multi | ✓ | 48.5 | 57.5 | 59.1 | 69.2 |
| ProHMR [25] | Multi | ✓ | 34.5 | N/A | 41.2 | N/A |
| Ours | Multi | ✓ | 33.0 | **34.4** | 41.6 | **46.4** |

Table 1: Comparison between the proposed method and existing single view and multiview methods on Human3.6M dataset. For each method, we specify the supported type of input and whether camera calibration is required. We evaluate the performance of each method on both multiview and single view reconstruction. Since multiview reconstruction is not supported by single view methods, we report the mean and standard deviation of the error after evaluating the method on each of the input views. In addition, we report the multiview methods (MV-SPIN [43] and LVS [43]) that use camera calibration for fusion as a reference, i.e., it is expected to provide the performance upper bound of our calibration-free method. MPJPE-PA and MPVPE-PA are reported in millimeter.

To our best knowledge, Liang [29] and ProHMR [25] are the only multiview methods that does not require known camera calibration.

**Multiview Reconstruction** We reconstruct 3D body shape and pose using multiview images from Human3.6M dataset as summarized in Table 1, middle three columns. For single view methods, we observe large variations in accuracy due to the varying viewing angle. Multiview methods overcome this limitation by fusing information from multiple views and producing a unified output. Figure 5 shows such a comparison between ours and SPIN [23] at continuous frames of a Human3.6M [15] sequence under MPVPE-PA. Compared to Liang [29] and ProHMR [25], our method outperforms in both MPJPE-PA and MPVPE-PA. We also report the performance of the multiview methods (MV-SPIN [43] and LVS [43]) that rely on the camera calibration as a reference, i.e., it is expected to provide the performance *upper bound* of our calibration-free method.

**Single View Reconstruction** Our method can be applied to an arbitrary number of cameras, including a monocular camera, *without* retraining the model. Applying our method on single view input, however, does not benefit from the attention-based multiview feature fusion. Evaluation results can be found in Table 1, last two columns. Even without specifically targeting for the single view use case, the results produced by our method is comparable with existing ones on MPJPE-PA (except MeshTransformer [30]) that use a transformer network as backbone) while outperform others on MPVPE-PA. Further, our fusion method is complementary to the single view reconstruction approaches where we can extract the global and local features with a minor modification. We expect that a stronger performance can be achieved in the multiview reconstruction if a stronger baseline can be used.

## 4.4 Ablation study

For all the following experiments, we train the model using the Human3.6M and UP-3D datasets, and evaluate the model on the test split of Human3.6M.

**Effectiveness of Local Features** We compare our model with the variant that is trained with the global features only. The features across views are fused with the self-attention mechanism. Table 2 shows that adding local features effectively improves 3D reconstruction accuracy.

**Effectiveness of using Self-attention** To demonstrate the effectiveness of our attention based fusion, we compare ours with other alternative fusion methods: average pooling and max pooling. Comparison in Table 2 demonstrates that the model trained with our self-attention fusion outperforms the other fusion methods by a large margin.

**Benefit of using More Views** To investigate the benefit of using more views for 3D reconstruction, we evaluate the performance of our model with varying number of views, randomly selected from all four views. Note that re-training the model is not necessary since our approach allows taking an arbitrary number of views as input. The results are summarized in Table 2. Using more views helps

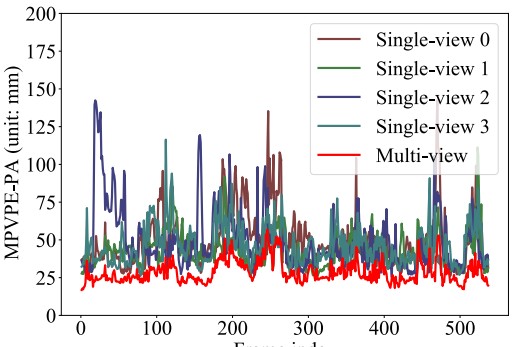

Figure 5: We compare ours that infer jointly from multiple views with SPIN [23] infer on each view independently on a sequence from Human3.6M [15]. Our multi-view results are equivalent to or better than single-view result from the best view for almost all frames.

| Variant | MPJPE-PA | MPVPE-PA |
|---|---|---|
| W/o local feature | 37.7 | 42.0 |
| Average fusion | 35.4 | 37.8 |
| Max fusion | 35.2 | 37.5 |
| Att. fusion (ours) | **33.0** | **34.4** |
| Att. fusion (3 views) | 34.2 | 36.6 |
| Att. fusion (2 views) | 37.3 | 40.2 |
| Att. fusion (1 view) | 44.1 | 50.1 |
| Att. fusion (1 view, front) | 41.5 | 46.1 |
| Att. fusion (1 view, back) | 46.5 | 53.2 |

Table 2: We compare different variants of the proposed method to verify the design decisions. All the variants are trained on the union of Human3.6M and UP-3D and evaluated on Human3.6M (use all four views unless specified). MPJPE-PA and MPVPE-PA are reported in millimeter.

improve the performance. Also, compared to frontal views (camera 1 and 3), back views (camera 0 and 2) that are more prone to self-occlusion benefit the most from information fused from other views.

### 4.5 Qualitative Results

We apply our method to reconstruct 3D body shape and pose using multiview images and show the qualitative comparisons on images from Human3.6M [15], VBR [2], MARCOnI [6] and self-collected Social Videos dataset against SPIN [23] in Figure 6.

### 4.6 Failure Cases

In Figure 7, we show qualitative results of failure cases including (1) failure of dense keypoints (example A and B) and (2) under severe occlusion or crowded (example C).

## 5 Summary and Future Work

We present a novel 3D human body reconstruction method that uses multiview images without calibration. Instead of relying on geometric camera calibration, our method leverages the observed human subject as a semantic calibration target that can align the visual features from multiview images. The visual features are learned in the 3D body surface coordinate, established by the dense keypoint mapping, that is agnostic to views and poses. We reason about self-occlusion via the confidence of the aligned features, which allows us to effectively fuse the features from multiview images. The fused features are fed into a MLP regressor to regress model parameters. Experiments show that our calibration-free method is able to effectively leverage multiview information in a principled way, outperforming state-of-the-art single view approaches with a post-hoc multiview fusion. Our approach is readily applicable to real world scenarios, including 3D reconstruction from social videos.

Our reliance on the dense keypoint estimation can be a double-edged sword. While it establishes the dense correspondences without camera calibration, the erroneous estimation of the dense keypoints may lead to misalignment of the local features, resulting in suboptimal 3D reconstruction. Our attention mechanism can mitigate fusing erroneous features if the majority of dense keypoint estimation from multiview images are correct. In practice, we found that the dense keypoint estimation [11] is highly reliable, which makes our method applicable to in-the-wild social videos.

We believe camera calibration, if available, provides additional point-to-line correspondences, which can be used to not only validate, but also refine the point-to-point correspondences our feature fusion method relies on. Exploring a combination of both, along with developing robust techniques to perform in-the-wild multiview calibration, is a promising future direction. Another potential future work is to extend our method to leverage temporal information and produce more temporally coherent predictions.

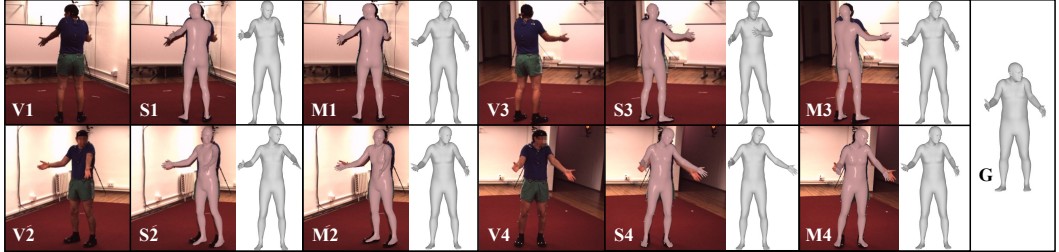

(a) Qualitative comparison with SPIN [23] on Human3.6M [15].

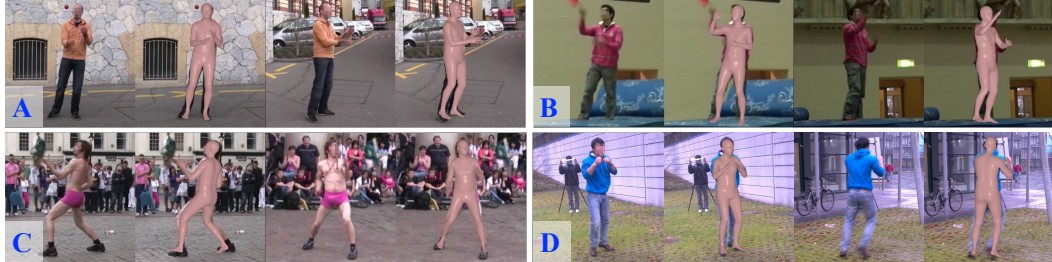

(b) Qualitative results on Unstructured VBR (A, B, C), and MARCOnI (D).

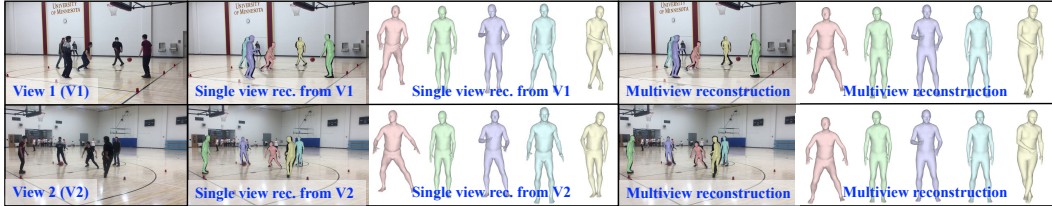

(c) Qualitative comparison with SPIN [23] on our Social Videos dataset.

Figure 6: (a) While reconstruction results of both may have similar level of alignment with image after reprojection, 3D reconstructions of single-view method are not consistent across views due to bias of viewpoint. "V" stands for "view", "S" stands for "Single-view reconstruction [23]", "M" stands for "Multiview reconstruction" of ours, "G" stands for "Ground truth". (b, c) We reconstruct 3D body shape and pose from uncalibrated multiview images of characteristic social videos in diverse scenes, including Unstructured VBR, MARCOnI, and our Social Videos. While single view reconstruction produces inconsistent poses depending on views, ours produces a consistent pose invariant to views as leveraging multiview information jointly.

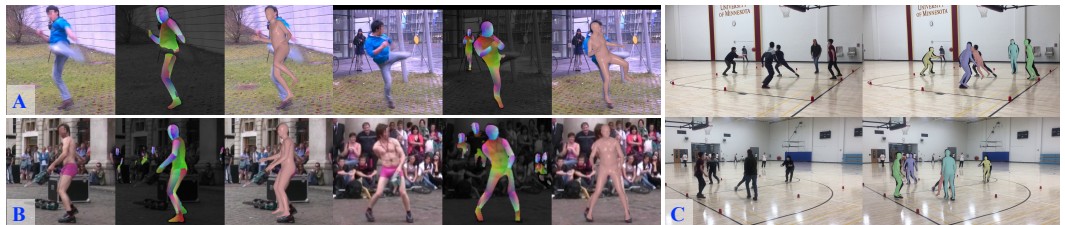

Figure 7: Failure cases. Dense keypoint estimation is missing for left leg in example A and right arm for example B for both views. In example C, subject rendered with reconstruction rendered red is under severe occlusion or crowded for both views.

## Acknowledgement

This work was partially supported by NSF NRI 2022894 and NSF CAREER 1846031.

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
