# Supplementary Material
# Multiview Human Body Reconstruction from Uncalibrated Cameras

**Zhixuan Yu**
University of Minnesota
yu000064@umn.edu

**Linguang Zhang**
Meta Reality Labs
linguang@meta.com

**Yuanlu Xu**
Meta Reality Labs Research
yuanluxu@meta.com

**Chengcheng Tang**
Meta Reality Labs
chengcheng.tang@meta.com

**Luan Tran**
Meta Reality Labs
tranluan07@meta.com

**Cem Keskin**
Meta Reality Labs
cemkeskin@meta.com

**Hyun Soo Park**
University of Minnesota
hspark@umn.edu

## A Experiment Details

For training, we mixed 312K images from the Human3.6M [5] (multi-view) and all 8.5K images from the UP-3D [12] (single-view). We randomly select samples from them with a ratio of 0.8 : 0.2 probabilistically for each batch. For each selected Human3.6M sample, we determine the number of views to use (1 - 4) uniformly and pick them randomly. Data samples without a corresponding ground truth (e.g. 3D joint positions) will have the corresponding loss term set to 0 in equation (9). We use a batch size of 16 and a learning rate of $3 \times 10^{-5}$. We set $\lambda_P = \lambda_K = 5.0$. For testing on Human3.6M, we use 110K images for multiview reconstruction and 27.5K images for single-view reconstruction.

## B Qualitative Comparison with Multiview Methods

We show some qualitative comparisons with another multiview method [15] and a single-view method [9] with post-hoc multiview fusion (average of independent predictions from each view) in Figure 1. Each method reconstruct a unified mesh from multiview inputs.

Our method makes better predictions for most cases, which shows the effectiveness of our multiview fusion strategy.

## C Quantitative Evaluation on In-the-wild Datasets

We perform additional quantitative evaluations on two challenging in-the-wild multiview dataset, Ski-Pose [20] and MannequinChallenge [14], for comparison with several baseline methods. Ski-Pose captures skiers performing giant slalom runs by six synchronized and calibrated pant-tile-zoom-cameras (PTZ). It comes with ground truth 3D pose and follow its standard split for training and testing. MannequinChallenge is captured by a moving hand-held camera while subjects stay still in daily life scenarios. We use annotations provided by Leroy *et al.*[13] following ProHMR [11].

36th Conference on Neural Information Processing Systems (NeurIPS 2022).

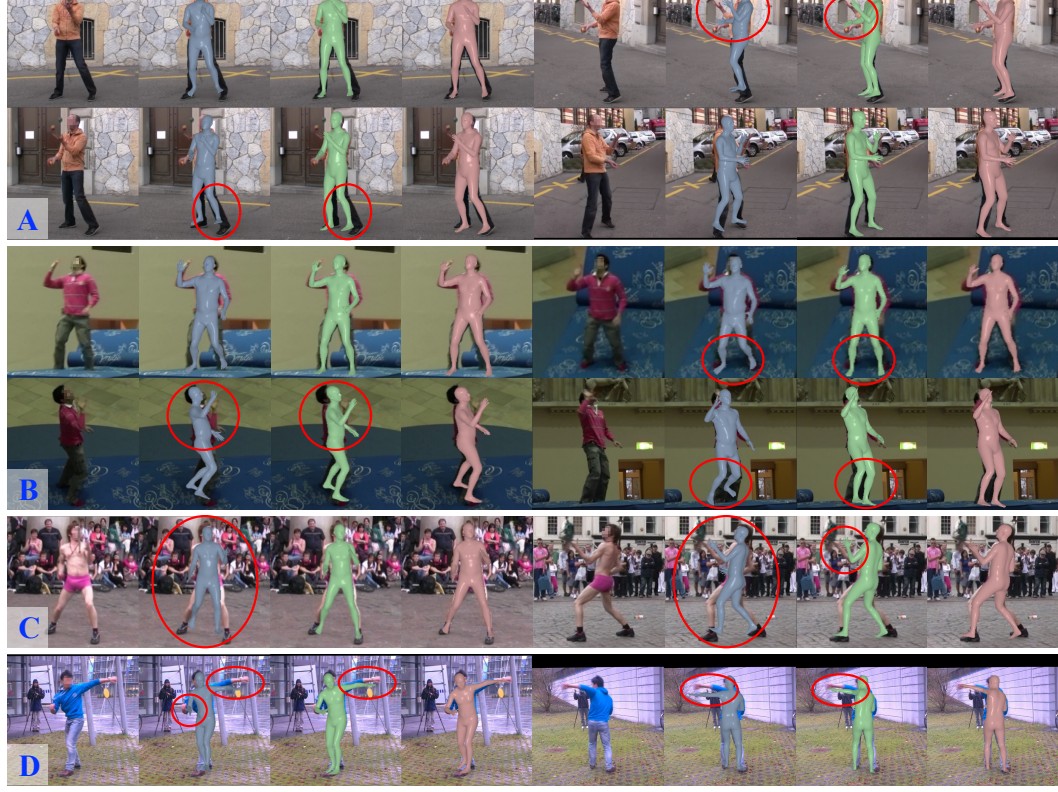

Figure 1: Qualitative comparison between Liang [15] (blue), post-hoc multiview fusion (average of body model parameters across views) of SPIN [9] (green) and ours (pink) on Unstructured VBR [1] (A, B, C), and MARCOnI [4] (D). Each block shows multiview images of the same subject taken at the same time and reprojections of the 3D mesh reconstructed by each method. Red circles highlight regions with wrong predictions.

| Method | Type | Calibration-free | Ski-Pose [20] | MannequinChallenge [14] |
|---|---|---|---|---|
| SPIN [9] | Mono | ✓ | $125.3 \pm 23.9$ | N/A |
| DecoMR [21] | Mono | ✓ | $166.3 \pm 40.3$ | N/A |
| Liang [15] | Multi | ✓ | 123.5 | 95.4 |
| ProHMR [11] | Multi | ✓ | 108.1 | 85.1 |
| Ours | Multi | ✓ | **93.0** | **83.7** |

Table 1: Additional comparison between the proposed method and existing single view and multiview methods on Ski-Pose [20] and MannequinChallenge [14] datasets. We report MPJPE-PA metric in millimeter.

We reconstruct 3D body shape and pose using multiview images from those datasets and report MPJPE-PA metric in Table 1.

## D   Model Complexity and Runtime

There are 45.3M learnable parameters in our architecture, among them 12.6M are used for the self-attention mechanism. The run times per frame compared with other methods and compared with different variants of our method are summarized in Table 2 (unit: ms). Note that we do not especially optimize for runtime efficiency. We test run time for a different number of views within a frame. Unlike other single-view methods, the run time of our method is not proportional to the number of views within a frame.

| Method | 1 view | 2 views | 3 views | 4 views | 5 views | 6 views |
|---|---|---|---|---|---|---|
| HMR [6] | 13.8 | 27.6 | 41.4 | 55.2 | 69.0 | 82.8 |
| PARE [8] | 14.8 | 29.6 | 44.4 | 59.2 | 74.0 | 88.8 |
| MeshTransformer [16] | 28.2 | 56.4 | 84.6 | 112.8 | 141.0 | 169.2 |
| PyMAF [22] | 30.0 | 60.0 | 90.0 | 120.0 | 150.0 | 180.0 |
| W/o local feature (Ours) | 18.3 | 26.9 | 37.8 | 45.1 | 55.8 | 64.4 |
| W/o self-Attention (Ours) | 35.5 | 46.2 | 56.3 | 65.8 | 75.0 | 86.4 |
| Ours | 36.0 | 46.9 | 57.4 | 66.9 | 76.3 | 87.6 |

Table 2: Inference time per frame. Unit: ms

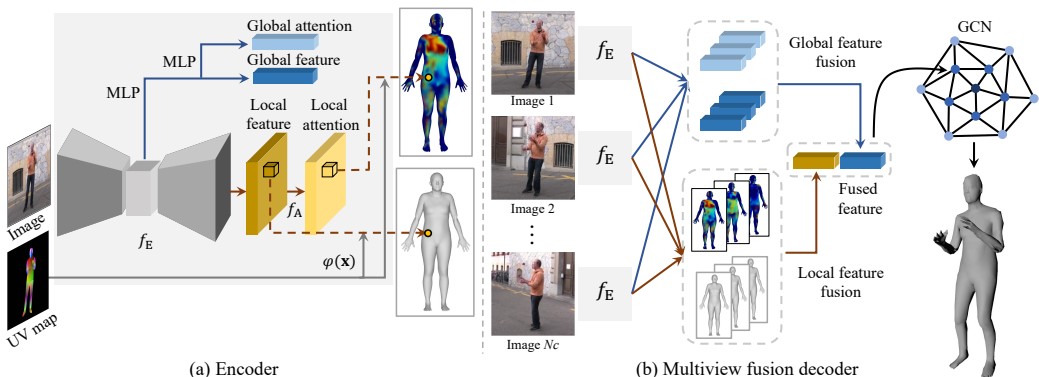

(a) Encoder        (b) Multiview fusion decoder

Figure 2: Network architecture with a graph convolutional network (GCN) as mesh decoder that learns to predict 3D locations of the mesh vertices as non-parametric representation of human body pose and shape.

## E   Graph Convolutional Network as an Alternative Mesh Decoder

Inspired by GraphCMR [10], we also explore directly regressing 3D locations of the mesh vertices via a graph convolutional network (GCN). The GCN is tailored to retain the SMPL template mesh topology [17] that learns the relationship between features and 3D vertex locations.

Specifically, we use a graph convolutional network (GCN) as the mesh decoder $f_{\mathrm{D}}(\mathbf{F}; \theta_D)$ in Equation (1) of Method section that reconstruct 3D body shape and pose from the set of the fused features, $\mathbf{F} = \{\bar{\mathbf{F}}_i^*\}_{i=1}^{N_v}$, where each node and edge is defined by the topology of the triangular mesh. Each hidden graph convolutional layer [7] is defined as:

$$Y = \tilde{A}XW \tag{1}$$

where $\tilde{A} \in R^{N \times N}$ is the row-normalized adjacency matrix, X is the input feature and $W$ is the weight matrix. Following GraphCMR [10], we downsample the mesh for the purpose of feature learning and upsample the reconstructed mesh to the original scale using pre-computed matrices to reduce redundancy in the original mesh. Each node is assigned with the concatenation of a corresponding local feature and the same copy of the global feature. In practice, we downsample SMPL vertices as the factor of four for feature learning and upsample the 3D reconstruction to the SMPL resolution [10]. The graph convolutional mesh decoder has 7 layers, each with 256 channels. Figure 2 show the updated network architecture with GCN as mesh decoder to predict non-parametric representation of human body pose and shape. We replace the loss for SMPL [17] model parameters with an L2 loss for vertex locations to train this network.

We conduct experiments for this alternative architecture and add results in Table 3 as the additional item "Ours (GCN)". While it outperforms GraphCMR [10] by a large margin due to the incorporation of local features and attention-based multiview fusion, it underperforms the model that regress parametric representation with MLP (i.e. "Ours") on all metrics.

**Discussion** We speculate the performance of this model variant that directly regress mesh vertices can be limited by the inevitable coupling between view-dependent (holistic orientation) and view-independent (pose and shape) components in the non-parametric representation. In the single-view setting that is adopted by many existing methods, the network can straightwardly predict non-parametric representation in the camera coordinate system, where the final reconstruction directly maps to visual cues extracted from the input view. In contrast, an uncalibrated multiview setting

| Method | Type | Multiview reconstruction | | | Single view reconstruction | |
|---|---|---|---|---|---|---|
| | | Calibration-free | MPJPE-PA | MPVPE-PA | MPJPE-PA | MPVPE-PA |
| SMPLify [2] | Mono | ✓ | N/A | N/A | 82.3 | N/A |
| HMR [6] | Mono | ✓ | 57.8 ± 10.7 | 67.7 ± 15.4 | 56.8 | 65.5 |
| GraphCMR [10] | Mono | ✓ | 50.9 ± 9.1 | 59.1 ± 13.4 | 50.1 | 56.9 |
| SPIN [9] | Mono | ✓ | 44.5 ± 7.9 | 51.5 ± 11.8 | 41.1 | 49.3 |
| DecoMR [21] | Mono | ✓ | 42.0 ± 8.8 | 50.5 ± 14.1 | 39.3 | 47.6 |
| Pose2Mesh [3] | Mono | ✓ | N/A | N/A | 47.0 | N/A |
| I2lMeshnet [18] | Mono | ✓ | N/A | N/A | 41.1 | N/A |
| MeshTransformer [16] | Mono | ✓ | N/A | N/A | **36.7** | N/A |
| PyMAF [22] | Mono | ✓ | N/A | N/A | 40.5 | N/A |
| MV-SPIN [19] | Multi | ✗ | 35.4 | N/A | N/A | N/A |
| LVS [19] | Multi | ✗ | **32.5** | N/A | N/A | N/A |
| Liang [15] | Multi | ✓ | 48.5 | 57.5 | 59.1 | 69.2 |
| ProHMR [11] | Multi | ✓ | 34.5 | N/A | 41.2 | N/A |
| Ours (GCN) | Multi | ✓ | 36.1 | 36.6 | 44.8 | 50.0 |
| Ours | Multi | ✓ | 33.0 | **34.4** | 41.6 | **46.4** |

Table 3: Experiment results of the proposed method with graph convolutional network (GCN) as mesh decoder added. Refer to Table 1 of main manuscript for detailed description.

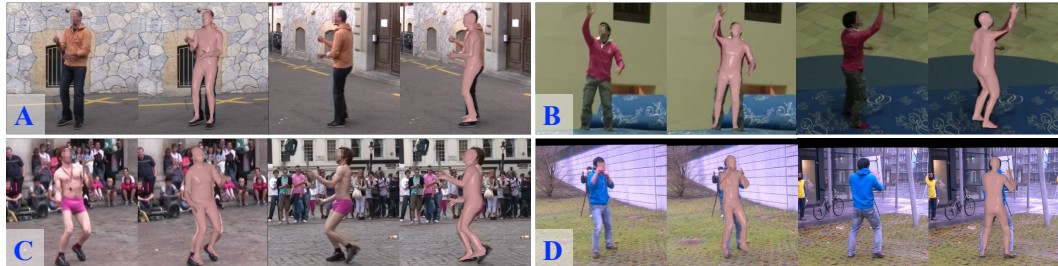

Figure 3: Qualitative results of the model variant using GCN as mesh decoder on Unstructured VBR [1] (A, B, C), and MARCOnI [4] (D).

forces the network to predict the non-parametric representation in a canonical coordinate system, which cannot directly leverage view-dependent visual cues. This can diminish the advantage of using non-parametric representation as regression target in multiview scenario.

We apply this model variants to reconstruct 3D body shape and pose using multiview images and show some example results in Figure 3

## F   Additional Qualitative Results

We show some additional qualitative results on our Social Video data in Figure 4.

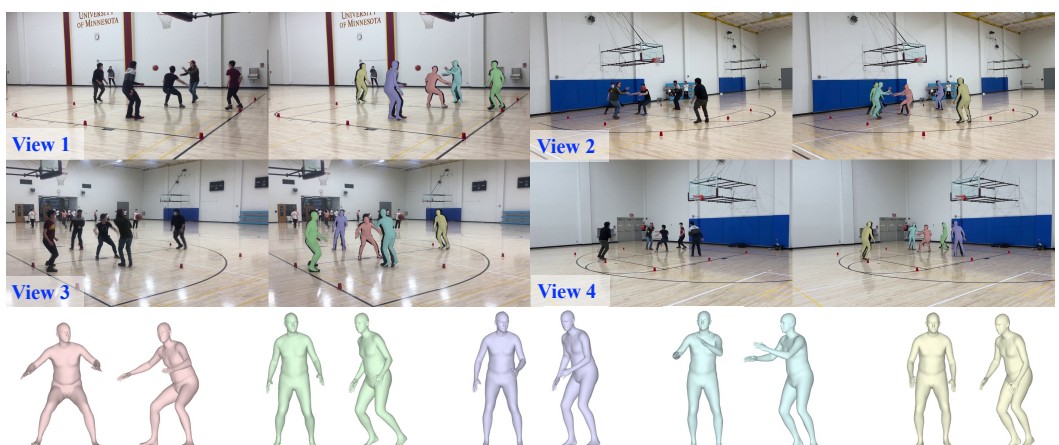

Figure 4: More qualitative results on our Social Videos. We show multiview reconstructions reprojected to each view as well as in 3D from both front view and side view. Same subject across views and in 3D are rendered with the same color.