# OpenReview forum: "Multiview Human Body Reconstruction from Uncalibrated Cameras"
_NeurIPS.cc/2022/Conference — NeurIPS 2022 Accept_

### Official Review · Reviewer_bCFQ · 2022-07-05

**Rating:** 7
**Confidence:** 3
**Soundness:** 4 excellent
**Presentation:** 2 fair
**Contribution:** 3 good

**Summary:**

This paper presents a new method for 3D human pose reconstruction give multiple (or a single) image without the need for camera parameters. The method is based on doing dense pose estimation, then predicting local and global features for how salient parts of an image are from a specific view. Finally it regresses parameters of a SMPL model which parameterizes the human body.

------------
Post rebuttal: Changing score from 6 to 7. I think the authors can improve the presentation of the paper and with that I would rate it a 7. I like the approach and design along with the motivation of using the human body as the semantic target.

**Questions:**

What's the reason why the final parameters were regressed instead of using an optimization based approach like SMPLify? It should be possible to do that without a calibration (optimize the 2D pixel loss from the output to the dense keypoint mapping) and it would ensure the output matches up with the visible images as much as possible.

It seems like the main proposed applications is for videos, and there is a large body of work on 3D pose from video. Is there a reason the authors chose to specifically focus on multi-view images without using temporal information?

**Limitations:**

Limitations are well addressed in the final section.

If the authors can address questions about the presentation and clarity, and include that in a revised version, then I would consider increasing my review score given that the contribution and method are interesting.

**Strengths And Weaknesses:**

Strengths:
The intuition behind the paper makes a lot of sense especially the way it was presented in terms of global features and local features with an attention map. I particularly liked the experiments section. There were detailed comparisons to other methods, and the ablation study did a good job of justifying the different design decisions. I would have liked to see a comparison with SMPLify in the table as well, but a number of other single view baselins were presented.

Weaknesses:
The presentation of the paper could be improved, especially the methods section. For example, figure 4 which shows the full method overview should appear near the top, especially before figure 3 and 2. Seeing the system overview is important before diving into the specific pieces. Figure 4 should also include $F_A$ so we can see where those weights are. It also was not clear how the weights for $F_A$ are learned. Eq. 7 has two unknowns: $a_c$ which is the attention score and $F_A$ which is the weights of the attention module. Do we have access to the ground truth attention score during training in order to learn $F_A$?

The other main weakness is that the results shown in the video still contain considerable artifacts especially in the way the hand moves. Compared to other methods for related 3D reconstruction tasks, such as Nerf based methods, the artifacts are very noticeable. The results are better than the single channel baseline, but could still be better especially given the number of camera views.

---

> ### Author Response · Authors · 2022-08-02
> **Rebuttal**
>
> Thank you for the constructive review. Below we try our best to address your concerns and questions:
>
> [W1] “The presentation of the paper could be improved, especially the methods section”
>
> We will move figure 4 (pipeline) on top of page 4 before figure 3 and 2 as well as mark $F_A$ in figure 4(1) for clarity. We do not need ground truth attention scores to learn $F_A$. Instead we only supervise the final predicted mesh, which implicitly encourages the attention function to assign higher scores to more informative views. An analogy is that the Spatial Transformer Network [1] learns affine transformation by supervising the class prediction only.
>
> [W2] “The other main weakness is that the results shown in the video still contain considerable artifacts especially in the way the hand moves…”
>
> Rendering and 3D reconstruction are different tasks. For a NeRF-based novel view synthesis method, plausible rendering without artifacts is the ultimate goal to pursue. Our work falls into the broad category of 3D reconstruction and focuses on reconstructed 3D geometry. This goes beyond novel view synthesis and requires semantically meaningful poses. Besides, we are aiming at sparse camera settings that often have considerably less number of views. Moreover, to make our method more applicable in the wild, a primary focus of the proposed method is to relax the dependency on camera calibration, which Nerf-based methods typically require as input.
>
> [Q1] “What's the reason why the final parameters were regressed instead of using an optimization based approach like SMPLify?”
>
> The proposed pipeline could be augmented with an optimization-based approach like SMPLify to refine the regression result. We are aware of its pros (better alignment with the visible images) and cons (higher computational cost). Since the work of SPIN [2], the advantage of combining a regression-based approach with an optimization-based approach has been widely recognized. However, because this post-processing method can be generalized to any regression-based method, a common practice for evaluation is to compare regressed results (even in SPIN [2]). We will add a comparison with SMPLify as suggested.
>
> An optimization-based method like SMPLify could also be applied as a post-processing step of aggregating single-view results. However, such a pipeline requires heuristics for estimating visibility of individual views, alignment of incomplete surfaces, and aggregation with semi-arbitrarily chosen weights, in contrast to an end-to-end learnable framework as ours.
>
> [Q2] “Is there a reason the authors chose to specifically focus on multi-view images without using temporal information?”
>
> We agree that extending our method to leverage temporal information could enable the method to produce more temporally coherent predictions. We regard this extension as future work because the proposed method focuses on using the human body as a semantic calibration target to assist multiview feature fusion, which is critical to reconstructing the human mesh under occlusion. In contrast, a temporal model could only hallucinate the occluded body parts given neighboring frames.
>
> Again, thank you for reviewing our paper and bringing up those meaningful questions and suggestions. We hope our response addresses your concerns and questions. Please let us know if you have more questions. Thanks!
>
> References
>
> [1] Jaderberg, et al. "Spatial transformer networks." NeurIPS. 2015.
>
> [2] Kolotouros, et al. "Learning to reconstruct 3D human pose and shape via model-fitting in the loop." ICCV. 2019.

---

### Official Review · Reviewer_w6Ne · 2022-07-09

**Rating:** 7
**Confidence:** 5
**Soundness:** 3 good
**Presentation:** 3 good
**Contribution:** 3 good

**Summary:**

This paper presents a calibration-free method to reconstruct 3D human body pose and shape from multiview images. The proposed method is motivated by the observations that the human body can be leveraged as a semantic calibration target. Experimental results validate the effectiveness of the proposed method. Both qualitative and quantitative results are promising.

**Questions:**

- The supplementary material provides a version using GCN to regress mesh vertices directly, but the quantitative results are counterintuitive as the single-view methods with non-parametric representations typically outperform the parametric counterparts numerically. But it makes sense that there are artifacts in the reconstructed meshes of the non-parametric solutions, as shown in Figure 3 of the supplementary material. It is also recommended to add ablation experiments that use the GCN mesh decoder when camera parameters are available.

- The related work could be improved to be more comprehensive.  It is recommended to include [A] in the related work, as it is a recent state-of-the-art multiview reconstruction method when camera calibration is available.

[A] Lightweight multi-person total motion capture using sparse multi-view cameras. ICCV 2021.


**Limitations:**

Please see the comments in Weaknesses and Questions.

**Strengths And Weaknesses:**

Generally, this paper is well-motivated, well-written, and technically sound. It is quite smooth to read this paper.

Overall, this paper presents an interesting solution for uncalibrated multi-view human pose and shape estimation.

+ The motivation of leveraging the human body as a semantic calibration target is appreciated.

+ The approximation of the discrete feature mapping via continuous mapping is interesting, which is the key to achieving differentiable feature learning. It is reminiscent of the existing differentiable rendering approaches.

Some concerns regarding the current version of the paper:

- Some part of the technical details is not very clear. It is confusing to me how to concatenate the per-pixel features with the global features?

- How about the proposed method when the camera calibration is available? The continuous mapping may also contribute to better fusion when the camera parameters are known. Adding such a setting in the ablation study should strengthen the paper.

---

> ### Author Response · Authors · 2022-08-02
> **Rebuttal**
>
> Thank you for the constructive review. Below we try our best to address your concerns and questions:
>
> [W1] “...how to concatenate the per-pixel features with the global features?”
> We will add the following implementation details: For the architecture with MLP as the mesh decoder, each local feature vector goes through a three-layer MLP and reduces its dimension from 256 to 5. Each of them is associated with a vertex of a down-sampled SMPL mesh [1] and we concatenate all 431 local feature vectors to form a 2155-dimensional feature vector. This aggregated local feature vector is then concatenated with the global feature vector and fed into a MLP regressor similar to that in HMR [2]. For the architecture with GCN as the mesh decoder, each local feature vector is concatenated with the same copy of the global feature and assigned to the corresponding GCN node.
>
> [W2] “How about the proposed method when the camera calibration is available?...”
>
> We are strongly motivated by the uncalibrated multiview setting because it is highly applicable in the wild where obtaining camera calibration is practically difficult. For example, we ran an off-the-shelf structure-from-motion system (COLMAP [3]) to solve the camera parameters on the VBR dataset but bundle adjustment could only converge on one sequence. With that said, we believe camera calibration, if available, provides additional point-to-line correspondences, which can be used to not only validate, but also refine the point-to-point correspondences our feature fusion method relies on. Exploring a combination of both, along with developing robust techniques to perform in-the-wild multiview calibration, is a promising future direction.
>
> [Q1] “The supplementary material provides a version using GCN…”
>
> We initially explored the GCN variant, but as we discussed in L28-36 in the supplementary material, the inevitable coupling between view-dependent (holistic orientation) and view-independent (pose and shape) components in the non-parametric representation limits its performance in the uncalibrated multiview setting. In the single-view setting that is adopted by many existing methods, the network can straightwardly predict non-parametric representation in the camera coordinate system, where the final reconstruction directly maps to visual cues extracted from the input view. In contrast, an uncalibrated multiview setting forces the network to predict the non-parametric representation in a canonical coordinate system, which cannot directly leverage view-dependent visual cues.
>
> [Q2] “The related work could be improved to be more comprehensive.”
>
> We will add more recent works including to improve the related work section:
> - Zhang, Yuxiang, Zhe Li, Liang An, Mengcheng Li, Tao Yu, and Yebin Liu. "Lightweight multi-person total motion capture using sparse multi-view cameras." ICCV. 2021.
> - Zheng, Yang, Ruizhi Shao, Yuxiang Zhang, Tao Yu, Zerong Zheng, Qionghai Dai, and Yebin Liu. "Deepmulticap: Performance capture of multiple characters using sparse multiview cameras." ICCV. 2021.
> - Kocabas, Muhammed, Chun-Hao P. Huang, Otmar Hilliges, and Michael J. Black. "PARE: Part attention regressor for 3D human body estimation." ICCV. 2021.
> - Kolotouros, Nikos, Georgios Pavlakos, Dinesh Jayaraman, and Kostas Daniilidis. "Probabilistic modeling for human mesh recovery." ICCV. 2021.
> - Lin, Kevin, Lijuan Wang, and Zicheng Liu. "End-to-end human pose and mesh reconstruction with transformers." CVPR. 2021.
>
> Again, thank you for reviewing our paper and bringing up those meaningful questions and suggestions. We hope our response addresses your concerns and questions. Please let us know if you have more questions. Thanks!
>
> References
>
> [1] Kolotouros, et al. "Convolutional mesh regression for single-image human shape reconstruction." CVPR. 2019.
>
> [2] Kanazawa, et al. "End-to-end recovery of human shape and pose." CVPR. 2018.
>
> [3] Schonberger, etal. "Structure-from-motion revisited." CVPR. 2016.

---

### Official Review · Reviewer_Lrh9 · 2022-07-11

**Rating:** 6
**Confidence:** 3
**Soundness:** 2 fair
**Presentation:** 2 fair
**Contribution:** 2 fair

**Summary:**

The paper describes human pose and reconstruction with a multi-view method that can handle up to an arbitrary number of uncalibrated camera views.
The proposed method maps visual features from different views to a canonical body surface to reconstruct a fused 3D SMPL parametric model.
Depending on each view, a learnable parameter assigns confidence to the fused features, playing the role of attention.

Human3.6M, UP-3D, MARCOnI and VBR are the datasets taken into account for collecting different views. Experiments are done solely on Human3.6M,  with the P1 protocol. Concerning the considered competitors, the proposed method shows the best performance on MPVPE-PA, both for single and multi-view approaches.  Qualitative results are also shown on Social Videos dataset.


**Questions:**

There is minimal information about the experiments. Essential facts are the number of images used and whether different runs have been made with a changing number of images for the multi-view testing.
Similarly, for training, the method can randomly choose images to construct a model from multi-views, what kind of protocol is used to choose images, and how many samples are collected, especially in training.

The authors suggest that multi-view is relevant for solving occlusions, and in their Conclusions the authors write that there are cases in which it cannot be solved. A comparison with other methods, such as “PARE: Part Attention Regressor for 3D Human Body Estimation” (which uses a weak perspective camera), and a discussion on the conditions under which the method fails to solve occlusions is relevant for a complete understanding of the method.

The proposed loss uses both 3D and 2D information and the SMPL parameters \Theta; given that recently, many methods do not require paired 2D-3D data and only use weak – supervision, would the proposed multi-view implant be able to extend to unpaired 2d-3d data, using the multi-view?
What about runtime and number of parameters:  is the model competitive in terms of computation time and size?


**Limitations:**

Limitations are discussed in particular about self-occlusions

**Strengths And Weaknesses:**

The merit of the paper is to devise a multiview method using only key features and mapping them to a canonical model, which should, in principle, solve self-occlusion problems.
The paper is well structured, and the images are good, though it is not so well written.

A relevant weakness is a choice of experimenting only on Human3.6M.
Why not also use 3DPW, which consists of 35515 RGB images of 7 subjects with paired ground-truth SMPL parameters.
Many recent contributions use 3DPW for testing.

Furthermore, to compare with the single view methods, the authors report the mean and standard deviation of the error of each method tested on several individual inputs.
The comparison might be unfair since a single view method does not take advantage of several views.

Despite multi-views being claimed to be relevant for occlusions, there are no ablation studies on this issue.
Recent contributions such as “End-to-End Human Pose and Mesh Reconstruction with Transformers” and “Probabilistic Modeling for Human Mesh Recovery” have better results on single images than the authors. The authors should compare their method with the most recent works.

---

> ### Author Response · Authors · 2022-08-02
> **Rebuttal**
>
>
>
> Thank you for the constructive review.
>
> [W1] “A relevant weakness…”
>
> 3DPW only contains single-view images and cannot be used to demonstrate semantic feature fusion, which is the core contribution of the proposed method. We perform additional quantitative evaluations on two challenging in-the-wild dataset, Ski-Pose [1],
>
> | Method    | MPJPE-PA   |
> |----|---|
> | SPIN      | 125.3±23.9 |
> | DecoMR    | 166.3±40.3 |
> | Liang | 123.5|
> | ProHMR [4]    | 108.1 |
> | Ours| **93.0**  |
>
>
> and MannequinChallenge [2]. For the latter we use annotations provided by Leroy et al [3] following ProHMR [4].
>
> | Method    | MPJPE-PA |
> |------|-----|
> | Liang     | 95.4     |
> | ProHMR [4] | 85.1     |
> | Ours      | **83.7**     |
>
>
> [W2] “...The comparison might be unfair...”
>
> We would like to ask for a more concrete answer from the reviewer about what would make a fair comparison with single-view methods.
>
> [W3] “Despite multi-views being claimed to be relevant for occlusions…”
>
> Below we add two additional rows to ablation study (table 2), one for frontal views and another for back views (often incurs self-occlusion) in the Human3.6M dataset. The latter that are prone to self-occlusion benefit the most from information fused from additional views.
>
> | Variant                     | MPJPE-PA  | MPVPE-PA  |
> |-----------------------------|-----------|-----------|
> | Att. fusion (1 view, front) | 41.5      | 46.1      |
> | Att. fusion (1 view, back)  | 46.5      | 53.2      |
>
> [W4] “The authors should compare…”
>
> Below we include several additional recent works in Table 1 for comparison.
>
>
> | Method             | Type  | Calibration-free | MPJPE-PA (MV) | MPVPE-PA (MV) | MPJPE-PA (SV) | MPVPE-PA (SV) |
> |--------------------|-------|------------------|---------------|---------------|---------------|---------------|
> | Pose2Mesh [6]       | Mono  | Yes | -  | - | 47.0 | -  |
> | I2lMeshnet [7]      | Mono  | Yes | -| -   | 41.1 | -  |
> | MeshTransformer[5] | Mono  | Yes | -  | -  | **36.7**  | -  |
> | ProHMR [4]          | Multi | Yes  | 34.5 | -  | 41.2 | - |
> | Ours | Multi | Yes | 33.0  | **34.4**    | 41.6  | **46.4**  |
>
> [Q1] “There is minimal information…”
>
> We will add the following:  For testing on Human3.6M, we use 110K images for multiview reconstruction and 27.5K images for single-view reconstruction. For training, we mixed 312K images from the Human3.6M dataset and all 8.5K images from the UP-3D dataset. We randomly select samples from them with a ratio of 0.8 : 0.2 probabilistically for each batch.
> [Q2] “…A comparison with other methods… is relevant for a complete understanding of the method.”
>
> We have included more experiments in the table under W3. We will also discuss failure modes with examples qualitatively corresponding to each of the following cases: (1) under severe occlusion, (2) failure of dense keypoints estimation, (3) out-of-domain pose and/or shape.
>
> [Q3] “...be able to extend to unpaired 2d-3d data, using the multi-view?...”
>
> We could use data with paired 2D-3D and unpaired 2D-3D for hybrid training. Specifically, given two-view images without paired 3D ground truth, we can formulate a loss term to penalize discrepancies between their predicted 3D body shape and pose.
>
> [Q4] “What about runtime and number of parameters…”
> There are 45.3M learnable parameters (12.6M for the self-attention mechanism). The run times per frame compared with other methods and compared with different variants of our method are summarized in the following table (unit: ms):
>
>
> | \# Views  | 1    | 2    | 3    | 4     | 5     | 6     |
> |---------------------------|------|------|------|-------|-------|-------|
> | HMR | 13.8 | 27.6 | 41.4 | 55.2  | 69.0  | 82.8  |
> | PARE[6]  | 14.8 | 29.6 | 44.4 | 59.2  | 74.0  | 88.8  |
> | MeshTransformer[7]| 28.2 | 56.4 | 84.6 | 112.8 | 141.0 | 169.2 |
> | PyMAF | 30.0 | 60.0 | 90.0 | 120.0 | 150.0 | 180.0 |
> | W/o local feature (Ours)  | 18.3 | 26.9 | 37.8 | 45.1  | 55.8  | 64.4  |
> | W/o self-Attention (Ours) | 35.5 | 46.2 | 56.3 | 65.8  | 75.0  | 86.4  |
> | Ours   | 36.0 | 46.9 | 57.4 | 66.9  | 76.3  | 87.6  |
>
>
> Please let us know if you have more questions. Thanks!
> References
>
> [1] Rhodin, et al. "Learning monocular 3d human pose estimation from multi-view images." CVPR. 2018.
>
> [2] Li, et al. "Learning the depths of moving people by watching frozen people." CVPR. 2019.
>
> [3] Leroy, et al. "SMPLy benchmarking 3d human pose estimation in the wild." 3DV. 2020.
>
> [4] Kolotouros, et al. "Probabilistic modeling for human mesh recovery." ICCV. 2021.
>
> [5] Lin, et al. "End-to-end human pose and mesh reconstruction with transformers." CVPR. 2021.
>
> [6] Choi, et al. "Pose2mesh: Graph convolutional network for 3d human pose and mesh recovery from a 2d human pose." ECCV, 2020.
>
> [7] Moon, et al. "I2l-meshnet: Image-to-lixel prediction network for accurate 3d human pose and mesh estimation from a single rgb image." ECCV, 2020.
>
> [8] Kocabas, et al. "PARE: Part attention regressor for 3D human body estimation." ICCV. 2021.

---

### Official Review · Reviewer_RhpY · 2022-07-11

**Rating:** 7
**Confidence:** 4
**Soundness:** 3 good
**Presentation:** 3 good
**Contribution:** 3 good

**Summary:**

A 3D human body reconstruction method using multiview images without calibration is presented. The method leverages semantic calibration to align visual features learned from multiview images by dense keypoints detection.

**Questions:**

- It is not clear how the global features and local features are used in this method. Could you add more details?
- The idea of leveraging multiple datasets with different GT is very attractive. Could the authors provide details on how the training is performed?


**Limitations:**

- Qualitative results are shown to validate the multiview methods advantageous over single view in terms of consistency across views. This point is obvious. Thus, more qualitative results might be more valuable.

- Semantic calibration idea is sound for multiview reconstruction. If the idea could be extended to single view, or video-based reconstruction, that will be more practically useful.

**Strengths And Weaknesses:**

Strengths:
- The idea of using semantic calibration for human body reconstruction is sound and novel. Leveraging 2D/3D keypoints detection as well as multiple datasets make it an interesting approach.
- Main concepts and motivations are well and clearly delivered.

Weaknesses:
- The application of multiview reconstruction requires that sequences are temporally aligned. Comparing the single view methods, this might limit the use cases of the proposed method.

---

> ### Author Response · Authors · 2022-08-02
> **Rebuttal**
>
> Thank you for the constructive review. Below we try our best to address your concerns and questions:
>
> [W1] “The application of multiview reconstruction requires that sequences are temporally aligned…”
>
> As pointed out, our system requires synchronized video input. Nowadays, multiple camera manufacturers including GoPro provide multi-camera sync software using audio signals, which can be readily applicable.
>
> [Q1] “It is not clear how the global features and local features are used in this method.”
>
> We will add the following implementation details: For the architecture with MLP as the mesh decoder, each local feature vector goes through a three-layer MLP and reduces its dimension from 256 to 5. Each of them is associated with a vertex of a down-sampled SMPL mesh [1] and we concatenate all 431 local feature vectors to form a 2155-dimensional feature vector. This aggregated local feature vector is then concatenated with the global feature vector and fed into a MLP regressor similar to that in HMR [2]. For the architecture with GCN as the mesh decoder, each local feature vector is concatenated with the same copy of the global feature and assigned to the corresponding GCN node.
>
> [Q2] “The idea of leveraging multiple datasets with different GT is very attractive.”
>
> We will add the following training details:  When mixing Human3.6M (multi-view) and UP-3D (single-view) for training, we randomly select samples from them with a ratio of 0.8 : 0.2 probabilistically for each batch. For each selected Human3.6M sample, we determine the number of views to use (1 - 4) uniformly and pick them randomly. Data samples without a corresponding ground truth (e.g. 3D joint positions) will have the corresponding loss term set to 0 in equation (9).
>
> [L1] “...more qualitative results might be more valuable.”
>
> We will add more qualitative results including examples demonstrating the failure modes corresponding to each of the following cases: (1) under severe occlusion, (2) failure of dense keypoints estimation, (3) out-of-domain pose and/or shape.
>
> [L2] “...If the idea could be extended to single view, or video-based reconstruction, that will be more practically useful.
>
> We appreciate the reviewer by suggesting great future work.
>
>
> Again, thank you for reviewing our paper and bringing up those meaningful questions and suggestions. We hope that this addresses your concerns and questions. Please let us know if you have more questions. Thanks!
>
> References
>
> [1] Kolotouros, et al. "Convolutional mesh regression for single-image human shape reconstruction." CVPR. 2019.
>
> [2] Kanazawa, et al. "End-to-end recovery of human shape and pose." CVPR. 2018.

---

### Official Review · Reviewer_XdJs · 2022-07-17

**Rating:** 6
**Confidence:** 3
**Soundness:** 2 fair
**Presentation:** 3 good
**Contribution:** 3 good

**Summary:**

The authors present a technique to estimate the underlying pose and shape of a body from multiview images without requiring calibration information. The technique uses the human body as a calibration target and performs fusion of local and global features via a self-attention mechanism. The authors demonstrate the efficacy of the approach on standard benchmarks.

**Questions:**

1. Did the authors try to replace the self-attention mechanism with an explicit calibration based fusion mechanism? It would be useful to understand the similarities and differences of the self-attention with such an explicit fusion mechanism beyond average and max pooling discussed in the paper.
2. Please perform a quantitative evaluation on the social videos dataset as the the method is best suited for such a setting. This would entail calculating the relative pose between multiview images for comparison to the other methods, but this is a critical contribution and the wider audience would benefit from an analysis on "in-the-wild" dataset.
3. How does the method compare in run-time to other comparable methods?Did the authors experiment with bulkier or leaner verions of the architecture? A variant that runs in real-time would be especially useful for social videos setting.
4. Why havent the authors evaluated against [34] which seems to be solving the problem in a more relaxed setting? The authors should show some examples with single camera videos which are multiview over time, albiet with pose changes. The behavior of the method in such a setting, along with associated failure modes would be useful to understand from an application perspective.

**Limitations:**

The authors haven't discussed the failure modes of the approach adequately. The social videos dataset presents a good opportunity to qualitatively demonstrate when the method fails as mentioned in the limitations section (e.g. under severe occlusion). The authors haven't discussed the performance of the method on single camera videos which have greater practical appeal. Also the possibility to futher improve the approach with calibration by adapting the self-attention mechanism warrants some discussion to guide future work.

**Strengths And Weaknesses:**

Strengths:
1. The idea of using the body as a calibration target is novel.
2. The method shows comparable accuracy to methods requiring calibration which is not always available.
2. The authors have collected a new dataset (Social Datasets) to qualitatively evaluate the proposed approach.

Weaknesses:
1. The approach is not sufficiently motivated. Such a multiview approach applies to a multi-camera rig setting, and calculating the calibration is straight-forward. The authors should provide a more compelling discussion on why the method is of practical importance?
2. The method stands to be most useful in the context of the Social videos dataset. However, no quantitative evaluations exist on the same.
3. The assumption in line "141" of "We assume the 3D body vertex index i are consistent across people, views, and poses" is a strong assumption and forms the basis of the approach. Some quantitative analysis or commentary on this assumption will be useful for wider adoption. Some failure cases should also be shown.
4. The self-attention mechansim proposed in the paper serves to remove dependence on calibration and is the main contribution of the paper. The authors could do a better analysis of the usefulness of self-attention mechanism.
5. The architecture has limited novelty. For e.g., the self-attention mechanism could be reformulated as a transformer based multi-headed attention module. It would be useful to understand if such a formulation is beneficial.
6. The complexity and run-time of the method is not discussed.

Post rebuttal:
The authors have adequately responded to my concerns. I would have liked to visually see the failure cases. I am slightly increasing the rating of the paper based on addressing of my concerns.

---

> ### Author Response · Authors · 2022-08-02
> **Rebuttal**
>
>
> Thank you for the constructive review!
>
> [W1]
>
> Multiple view videos are prevalent in social media, e.g., social videos. The majority of these videos form a wide baseline system of cameras where both intrinsic and extrinsic calibration is infeasible in practice due to lack of visual correspondences. In fact, the publicly available multiview datasets including VBR are a few exceptions that provide camera parameters. To further validate our claim, we conducted a set of experiments to show infeasibility of camera calibration for wide baseline cameras. We used COLMAP [1], a structure-from-motion software to calibrate our Social video dataset as well as the VBR dataset. As expected, the bundle adjustment fails to converge and it reports that no good initial image pair was found, i.e., lack of correspondences. We will add the above discussion to motivate our work.
>
> [W2] [Q2]
>
> Obtaining 3D ground truth of human geometry from in-the-wild social videos is not trivial because no motion capture used in Human3.6M can be used. Instead, we perform quantitative evaluation on the MannequinChallenge [2] dataset using annotations produced by Leroy et. al. [3] following ProHMR [4]. This is a challenging in-the-wild dataset dedicated to hand-held multiview settings in daily life. Results are shown in the table below:
>
> | Method    | MPJPE-PA |
> |-----------|----------|
> | Liang     | 95.4     |
> | ProHMR [4] | 85.1     |
> | Ours      | **83.7**     |
>
> [W3]
>
> It is a strong yet effective assumption. In prior literature, SMPL [5] is shown to be an expressive model that can generate a diverse range of shapes and poses of humans. We used such a model to make semantic correspondences between wide baseline cameras where the requirement of calibration can be relaxed. Failure cases include misestimation of correspondences, which will be added in the paper.
>
> [W4]
>
> For clarification, our approach leverages “Human Body as Semantic Calibration Target” (Sec 3.1) and “Local Feature Registration” (Sec 3.2)  to remove dependency on calibration. The self-attention mechanism (Section 3.3) serves to fuse local features from multiple views in a more adaptive way (L163-166) that gives some performance boost over naive average/max fusion (Table 2).
>
> [W5]
>
> Our main novelty is the idea of using the human body as a semantic calibration target that enables fusing information from uncalibrated images in a principled way. The design of the self-attention module is not particularly new, but our secondary contribution is to integrate self-attention into the multiview fusion pipeline.
>
> [W6] [Q3]
>
> We conducted a new experiment to show computational complexity. There are 45.3M learnable parameters in our architecture, among them 12.6M are used for the self-attention mechanism. The run times per frame compared with other methods and compared with different variants of our method  are summarized in the following table (unit: ms):
>
>
> | \# Views                  | 1    | 2    | 3    | 4     | 5     | 6     |
> |---------------------------|------|------|------|-------|-------|-------|
> | HMR                       | 13.8 | 27.6 | 41.4 | 55.2  | 69.0  | 82.8  |
> | PARE[6]                   | 14.8 | 29.6 | 44.4 | 59.2  | 74.0  | 88.8  |
> | MeshTransformer[7]        | 28.2 | 56.4 | 84.6 | 112.8 | 141.0 | 169.2 |
> | PyMAF                     | 30.0 | 60.0 | 90.0 | 120.0 | 150.0 | 180.0 |
> | W/o local feature (Ours)  | 18.3 | 26.9 | 37.8 | 45.1  | 55.8  | 64.4  |
> | W/o self-Attention (Ours) | 35.5 | 46.2 | 56.3 | 65.8  | 75.0  | 86.4  |
> | Ours                      | 36.0 | 46.9 | 57.4 | 66.9  | 76.3  | 87.6  |
>
>
> Note that we do not especially optimize for runtime efficiency. We test run time for a different number of views within a frame. Unlike other single-view methods, the run time of our method is not proportional to the number of views within a frame.
>
> [Q1]
>
> As we mentioned above, the calibration based methods do not apply because calibration is not feasible for wide baseline cameras, and therefore, it is not an option. Please also see our response in W1 regarding alternative approaches replacing our pipeline by calibrating cameras across images through bundle adjustment and the associated failures.
>
> [Q4]
>
> [34] is to recover shape, and therefore, it does not solve our problem: estimating shape and pose.
>
> Comments on Limitations
>
> - We will add the qualitative results of failure cases including (1) under severe occlusion, (2) failure of dense keypoints estimation, (3) out-of-domain pose and/or shape.
> - As we respond in Question 4 above, we believe the setting of a single camera moving over time as multiview input is out of the scope of the proposed method. Also, as we respond to W1, our target setting itself motivates the proposed work.
> - Due to character limit, please refer to our response for [W2] of Reviewer w6Ne.
>
> Please let us know if you have more questions. Thanks!

---

### Public Comment · ~Xiaoben_Li1 · 2023-08-21
**About the camera parameters**

Thanks for your wonderful work!

I am wondering that how you produce and represent camera parameters. When 2D reprojection loss is used, camera parameters of multiple views should be available, right?

Hope to get your reply!

---

> ### Public Comment · ~Zhixuan_Yu1 · 2023-08-28
> **Response to the question regarding camera parameters**
>
> Hi Xiaoben, thanks for your interest in our work and the question. We employ the weak-perspective camera model and the camera parameters are represented as (1) the global rotation in axis-angle representation, (2) translation and (3) scale. This follows HMR paper ("Kanazawa, Angjoo, Michael J. Black, David W. Jacobs, and Jitendra Malik. "End-to-end recovery of human shape and pose." CVPR. 2018.") and you can see details from sec 3.1 of that paper. Those parameters are predicted from the bottleneck feature of the feature encoder for each view separately.

---

> > ### Public Comment · ~Xiaoben_Li1 · 2023-09-22
> > **About the body pose and shape and camera parameters, as well as the evaluation of 3D pose.**
> >
> > Thanks a lot for your reply!
> >
> > I still have some question. You mentioned that the camera parameters are predicted for each view separately, then how about the body pose and shape parameters? In my understanding, the body pose and shape parameters of multi-view images of the same person should be identical, right? And for the evaluation, the 3D pose error of multi-view input is calculated in camera frame of each view and then averaged or calculated in other manner?
> >
> > Thanks again for your reply and look forward to more information.

---

> > > ### Public Comment · ~Zhixuan_Yu1 · 2023-10-10
> > > **Response to the question about the body pose and shape and camera parameters, as well as the evaluation of 3D pose**
> > >
> > > Hi Xiaoben, we predict unified body pose and shape parameters across views, and for the evaluation under multi-view input the 3D pose error is calculated in the canonical frame.

---

### Meta-Review · Area_Chair_ka5T · 2022-08-23

**Recommendation:** Accept
**Confidence:** Certain

**Metareview:**

This paper presents a calibration-free approach (as traditionally understood) for 3D modeling of humans using multi-view images. The idea of using the human body as a means of providing "semantic calibration" is interesting. The method is scalable with the number of cameras. The paper received three 7s and two 6s. The authors and reviewers had several engaging conversations. The reviewers have also provided several suggestions for improving the presentation of the paper.

**Award:**

No

---

### Decision · Program_Chairs · 2022-09-14

Accept